# A comprehensive SARS-CoV-2 genomic analysis identifies potential targets for drug repurposing

Nithishwer Mouroug Anand[1]☯, Devang Haresh Liya[1]☯, Arpit Kumar Pradhan[2,3]*, Nitish Tayal[4], Abhinav Bansal[5], Sainitin Donakonda[6], Ashwin Kumar Jainarayanan[7,8]*

1 Department of Physical Sciences, Indian Institute of Science Education and Research, Mohali, India,
2 Graduate School of Systemic Neuroscience, Ludwig Maximilian University of Munich, Munich, Germany,
3 Klinikum rechts der Isar, Technische Universität München, München, Germany, 4 Department of Biological Sciences, Indian Institute of Science Education and Research, Mohali, India, 5 Department of Chemical Sciences, Indian Institute of Science Education and Research, Mohali, India, 6 Institute of Molecular Immunology and Experimental Oncology, Klinikum rechts der Isar, Technische Universität München, München, Germany, 7 The Kennedy Institute of Rheumatology, University of Oxford, Oxford, United Kingdom, 8 Interdisciplinary Bioscience DTP, University of Oxford, Oxford, United Kingdom

☯ These authors contributed equally to this work.
* Arpit.Pradhan@campus.lmu.de (AKP); ashwin.jainarayanan@dtc.ox.ac.uk (AKJ)

**Data Availability Statement:** All relevant data are within the manuscript and its Supporting Information files.

## Abstract

The severe acute respiratory syndrome coronavirus 2 (SARS-CoV-2) which is a novel human coronavirus strain (HCoV) was initially reported in December 2019 in Wuhan City, China. This acute infection caused pneumonia-like symptoms and other respiratory tract illness. Its higher transmission and infection rate has successfully enabled it to have a global spread over a matter of small time. One of the major concerns involving the SARS-COV-2 is the mutation rate, which enhances the virus evolution and genome variability, thereby making the design of therapeutics difficult. In this study, we identified the most common haplotypes from the haplotype network. The conserved genes and population level variants were analysed. Non-Structural Protein 10 (NSP10), Nucleoprotein, Papain-like protease (Plpro or NSP3) and 3-Chymotrypsin like protease (3CLpro or NSP5), which were conserved at the highest threshold, were used as drug targets for molecular dynamics simulations. Darifenacin, Nebivolol, Bictegravir, Alvimopan and Irbesartan are among the potential drugs, which are suggested for further pre-clinical and clinical trials. This particular study provides a comprehensive targeting of the conserved genes. We also identified the mutation frequencies across the viral genome.

## Introduction

The 2019 novel coronavirus strain (2019-nCoV, later officially named SARS-CoV-2) which was initially reported in Wuhan, Hubei Province, People's Republic of China (PRC) belongs to the coronaviridae family of viruses that possess a positive-sense single-stranded RNA genome [1, 2]. Compared to the previous outbreaks of severe acute respiratory syndrome coronavirus (SARS-CoV) in 2003 and Middle East respiratory syndrome coronavirus (MERS-CoV) in

**Funding:** The author(s) received no specific funding for this work.

**Competing interests:** The authors have declared that no competing interests exist.

2012, 2019-nCoV has higher transmission and infection rate with an increasing mortality rate [3]. The SARS-CoV-2 genome like other members of the betacoronavirus family has a long ORF1ab polyprotein at the 5′ end, which is followed by a set of four major structural proteins, including the spike surface glycoprotein, small envelope protein, matrix protein, and nucleo-capsid protein (Fig 1) [4]. The 2019-nCoV strain and SARS-CoV share a genome sequence homology of about 79%. The 2019-nCoV has a greater similarity to the SARS-like bat CoVs (MG772933) than the SARS-CoV [1]. The high similarity of receptor-binding domain (RBD) in Spike-protein and several other analyses reveals that SARS-CoV-2 uses angiotensin-converting enzyme 2 (ACE2) as receptor, just like SARS-CoV. Coronavirus via the S protein on the surface identifies the corresponding receptor on the target cell thereby making its entry into the host cell [5]. The higher transmissibility and infection rate of 2019-nCoV as compared to SARS-CoV is attributed to the higher binding affinity of SARS-CoV-2 to the ACE2 receptors [6, 7]. In one of the structure model analysis, SARS-CoV-2 showed a 10-fold higher binding affinity for ACE-2 as compared to that of SARS-CoV [7]. The similarity of sequences between SARS-CoV-2 and SARS-CoV allows utilization of the known protein structures to build a model for drug discovery on this new SARS-CoV-2. A comprehensive genomic study could identify the start of community spread immediately and could help in imposing restrictions that could prevent subsequent infections [8].

As of January 23, 2021, total of 99,298,747 cases of COVID-19 occurring in at least 219 countries and territories were reported, with approximately 3% of fatality rate. The coronavirus similar to other RNA viruses is characterized by significant genetic variability and high recombination rate which boosts them to be easily distributed among humans and animals in different geographic locations [9]. Numerous coronavirus strains exist within the human and animal populations without causing life threatening diseases [10]. However in certain rare cases there is genetic recombination of viruses which produces infectious strains which are pathogenic to humans [11]. What makes SARS-CoV-2 more powerful is the mutation events that allow structural changes in the virus. One of the recent studies suggests the existence of three central variants of SARS-CoV-2 distinguished by amino acid changes [12]. There have been many studies which have performed phylogenetic analysis on SARS-CoV-2 genomes sampled from across the world. These studies have detailed the role of founder effects, genetics, immunological and environmental factors playing a confounding role in the evolution of SARS-CoV-2. These studies have identified several core mutations on the viral genome which have been linking them to the COVID-19 transition events [12–15]. With the increasing spread of the virus, there is an increase in the accumulation of mutation, which would thereby make pharmaceutical interventions difficult. We urgently need therapeutic options to combat this virus infection.

In this study, we thereby performed wide array analysis, which addresses the mutation problem and systematically identified drug targets to aid the therapeutic design. Firstly, we

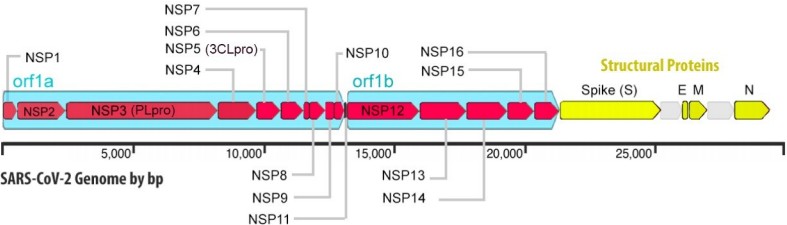

**Fig 1. A detailed schematic representation of the SARS-CoV-2 viral genome.** The figure represents the detailed view of structural and non-structural proteins (NSPs).

performed haplotype analysis, which identified several different primary clusters based on the haplotype network suggesting the presence of different variants of SARS-CoV-2. We also found the genes that are conserved and the population level variants. In this study, we also highlight the mutation frequencies across the viral genome. We then identified the stable genes, which have stretches of conserved regions and thereby can be used as efficient drug-targets. Using this as our base, we identified 4 genes which are stable and conserved in all the strains. We used them as our targets in in-silico drug designing, molecular docking and molecular dynamics simulations. Given the fast mutation rate of these viruses, our approach of targeting the stable genes through small molecules would provide a better therapeutic approach and confidence in the successive clinical trials. This study provides new insights into the evolution of COVID-19, identifies the divergence pattern, spread of the virus at the population level, and utilises a unique and efficient method of targeting the stable genes for the drug discovery approach.

## Results and discussions

### Viral clusters identified via haplotype network

In order to understand the population level divergence of SARS-CoV-2 we tried to map the haplotype network and establish the relationship among the SARS-CoV-2 haplotypes from the genome data collected all over the globe. A total of 194 haplotypes were identified from 358 SARS-CoV-2 genomes. Haplotype 1 had the highest prevalence and was present in diverse geographical locations (Fig 2). The main central hub consists of around 40–45% contribution from China followed by USA and Europe. However, the haplotype from the USA remains mostly inside the USA. The haplotype from China and Europe spread everywhere indicating more connectivity of these 2 regions with the rest of the world. This haplotype network may be incomplete because of the origin of the sequences from specific regions.

### Identification of conserved genes and mutation frequency across viral genome

To determine conserved regions we performed systematic sequence analysis, which identified the conserved genes with different threshold conservation levels (Table 1). The population variant genes were also identified and highlighted based on their geographic distribution (Table 2). Nucleotide positions 240, 3036, 8781, 11082, 14407, 23402, 28143, with reference to NC_045512.2 sequence, had mutation frequencies greater than 40 (Fig 3). This represents the highly mutating positions in the genome, which we call the "Hotspot Zones". These hotspot zones were distributed over the viral genome. Some of these zones lie in the NSP1, NSP3, NSP4, NSP6, NSP12, spike protein (S-protein) and ORF8 genes. For our further analysis, we chose the proteins with the highest conservation thresholds. NSP10, Nucleoprotein, PLpro, and 3CLpro were conserved targets, which were chosen for drug targeting. Interestingly, Japan had the least number of variant genes whereas in Asia the population carried a diverse set of SNPs throughout the viral genome (Table 2). Similarly, China, Rest of America (Mexico, Chile, Brazil) and Europe had more number of variant genes as compared to other populations in the UK and North America. Orf1a polyprotein was found to be a variant in all the population (Table 2).

### Homology modelling of stable targets and virtual screening of small molecules

The three dimensional structure generated by SWISS-MODEL was checked for its quality based on several parameters (Table 3). For each of the proteins, the models were arranged with

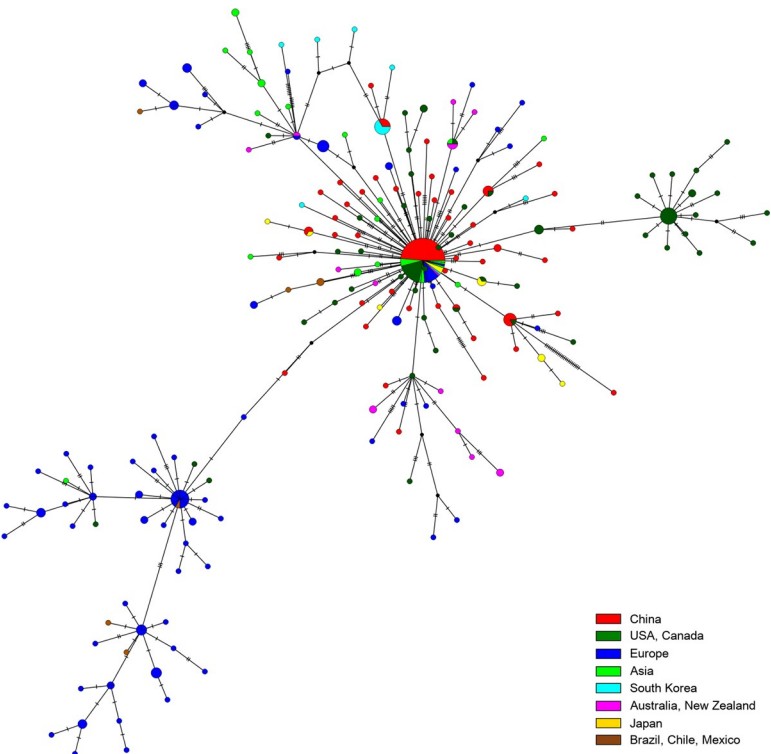

**Fig 2. Haplotype analysis of SARS-CoV-2 viruses.** Haplotype network of 358 SARS-CoV-2 viral genomes. The distribution of haplotypes over geographical areas were inserted as a part of the traits section in the Nexus file. The color code and its respective geographical distribution is marked on the bottom right corner.

respect to the GMQE scoring functions and were checked for the local quality estimate and Z-scores. The protein models, which were best fit in all these parameters, were assessed further for their quality (S1–S4 Figs).

All the four protein models had a greater proportion of residues in the favoured and allowed region in the Ramachandran Plot. PROSA Analysis revealed that structures are in the X-Ray/NMR structure fold and have a greater stereo chemical quality (S1–S4 Figs).

We used a Structure-Based drug designing and docking approach. We carried out the virtual screening of the drugs from the list of FDA approved drugs. MetaPocket 2.0 metaserver was used to identify the ligand-binding site on the protein surface. A binding site radius of 10 Å was defined and the docking was performed. The drugs, which docked to the proteins with a

**Table 1. Detailed list of conserved genes arranged into their respective thresholds of conservation.**

| THRESHOLD | THRESHOLD | THRESHOLD | THRESHOLD | THRESHOLD |
|---|---|---|---|---|
| **100–95** | **95–90** | **90–85** | **85–80** | **80–75** |
| Chain B, NSP10 | Chain A, Nucleocapsid protein | ORF1a polyprotein | ORF1ab polyprotein | ORF1ab polyprotein, partial |
| NSP10 | Chain A, Papain-like proteinase | Nucleocapsid phosphoprotein | ORF1a polyprotein, partial | Surface glycoprotein |
| Membrane glycoprotein | 3C-like proteinase | NSP2 | NSP3 | |
| Nucleocapsid phosphoprotein, partial | | ORF1ab polyprotein | NSP3 (residues 207–377) | |
| RNA binding domain of nucleocapsid protein | | | ADRP | |

**Table 2. Population wise variant genes arranged in reference to their geographical locations.**

| Oceania | China | Rest of America | UK | Japan | North America | Europe |
|---|---|---|---|---|---|---|
| NSP3 | ORF1a polyprotein, partial | ORF1a polyprotein, partial | ORF1ab polyprotein, partial | ORF1a polyprotein, partial | ORF1a polyprotein, partial | NSP13-pp1ab |
| ORF1a polyprotein, partial | ORF1a polyprotein | ORF1ab polyprotein | ORF1ab polyprotein | ORF1a polyprotein | NSP3 | Chain A, Uridylate-specific endoribonuclease |
| ORF1ab polyprotein | Chain A, Uridylate-specific endoribonuclease | ORF1a polyprotein | NSP2 | ORF10 protein | Chain A, Papain-like proteinase | NSP15-pp1ab (endoRNAse) |
| ORF1a polyprotein | Chain A, Replicase polyprotein 1ab | ORF1ab polyprotein, partial | ORF1a polyprotein | ORF10 protein, partial | Chain A, Peptidase C16 | ORF3a protein |
| NSP4 | Chain A, Non-structural Protein 3 | ORF1ab polyprotein | Surface glycoprotein | | ORF1a polyprotein | Membrane glycoprotein, partial |
| Spike glycoprotein | Chain A, NSP3 macrodomain | ORF3a, partial | Nucleocapsid phosphoprotein, partial | | ORF1ab polyprotein partial | Membrane glycoprotein |
| ORF3a protein | NSP3 | ORF3a protein | Chain A, Nucleoprotein | | ORF10 protein | ORF8 protein |
| Surface glycoprotein | ORF10 protein | Nucleocapsid phosphoprotein, partial | Chain A, SARS-CoV-2 nucleocapsid protein | | ORF10 protein, partial | ORF1ab polyprotein |
| Surface glycoprotein, partial | ORF10 protein, partial | Nucleocapsid phosphoprotein | NSP2 | | Chain A, SARS-CoV-2 NSP16 | NSP2 |
| ORF8 protein, partial | NSP14 | | | | Chain A, 2'-O-methyltransferase | ORF1a polyprotein, partial |
| ORF8 protein | | | | | | |
| ORF10 protein | | | | | | |

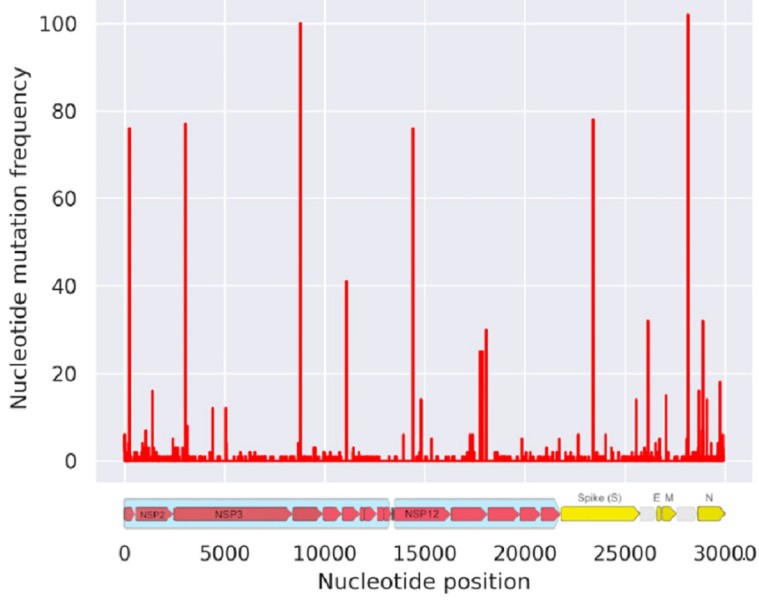

**Fig 3. Mutation frequency across the SARS-CoV-2 viral genome.** The red lines represent the number of mutations at a particular nucleotide position. On the abscissa is the nucleotide numbered from 0 to 30,000. To better understand the mutations across the viral genome, the genomic representation of SARS-CoV-2 is provided in the bottom panel. The red ones in the bottom panel represent the non-structural proteins while the yellow ones represent spike, E-proteins and the N-proteins.

**Table 3. Parameters for the validation of the homology modeled protein.**

| Proteins | GMQE Score | Q-Mean | Z-Score |
|---|---|---|---|
| PL-PRO | 0.11 | -0.28 | -8.87 |
| Nucleoprotein | 0.24 | 0.03 | -5.03 |
| NSP10 | 0.86 | -0.93 | -3.58 |
| 3CLPro | 0.99 | 0.45 | -7.2 |

higher docking score, were considered for the further analysis. For each protein, two drugs with highest docking scores were selected and were analysed further for the MD simulations. The docking scores for the best two drugs for each protein were: Nucleoprotein (Nebivolol: 83.7% Bictegravir: 83.5%), 3CL Pro (Nebivolol: 81.8 Darifenacin: 81.7) and NSP 10–16 Complex (Alvimopan: 81.8 Irbesartan: 80.8). The interaction of the drugs with the protein residues is visualised in Figs 4 and 5. Taken together, our structure based approach identified good quality models of stable proteins in SARS-CoV-2 and potential small molecules against them.

## Molecular dynamics (MD) simulation

Molecular Dynamics simulations are employed to study the strength and properties of the protein-drug complexes and their conformational changes on an atomic level. Various parameters such as RMSD, RMSF, Radius of Gyration, Intermolecular H-bonds, and SASA were calculated throughout the simulation trajectory to give insights on the structure of the proteins. To illustrate the dynamics, and conformational stability of the protein-drug complexes, the protein-drug complexes were subjected to MD simulations for a period of 100ns. The binding of the drugs Cilostazol and Elvitegravir destabilized the PLpro complex. Thereby Plpro was not short-listed for further downstream analysis. There were several interactions of Bictegravir and Nebivolol with the Nucleoprotein complex (Nucleoprotein-Bictegravir: Arg68, Gly124, Asn126; Nucleoprotein-Nebivolol: Pro67, Arg68, Tyr123, Ile131, Val133, Ala134). Alvimopan interacted with NSP10 at residues Asp82, His83, Phe89, Cys90, and Lys93 whereas Irbesartan had interactions with NSP10 at Cys74, His83, Pro84, Cys90, Leu92 and Leu112. While Darifenacin has some contacts with 3CLpro at Asn142, Asn214, Val303, Phe305, Nebivolol interacted with the 3CLpro at Lys751 and Thr763 residues (Figs 4 and 5). The binding site of Alvimopan in the NSP 10–16 complex was at the junction of both the protein complexes.

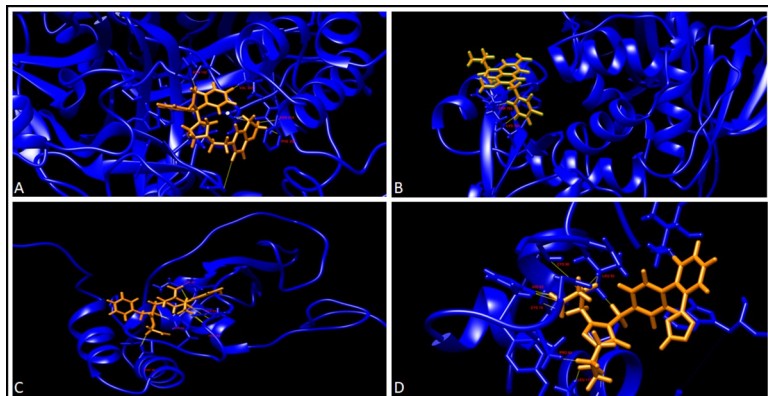

**Fig 4. Drug-protein interaction after docking.** A. 3CLPro-Darifenacin interaction, B. 3CLPro-Nebivolol interaction, C. NSP10-Alvimopan interaction, and D. NSP10-Isbesartan interaction. Drugs are in orange while the proteins are labelled in blue and the residues interacting with the drugs are highlighted in red. The contacts are shown in yellow.

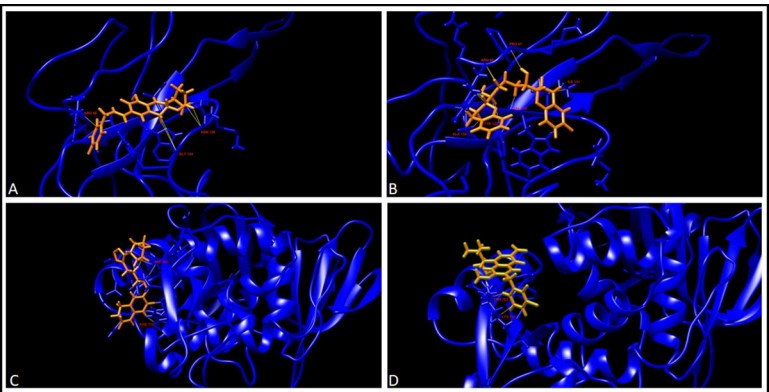

**Fig 5. Drug-protein interaction after docking.** A. Nucleoprotein-Bictegravir interaction, B. Nucleoprotein-Nebivolol interaction, C. PL Pro-Cilostazol interaction, and D. PL Pro-Elvitegravir interaction. Drugs are in orange while the proteins are labelled in blue and the residues interacting with the drugs are highlighted in red. The contacts are shown in yellow.

Alvimopan interacts with residues from both NSP 10 and NSP 16 complex (Thr 110 from NSP 16 and with Lys 93 from NSP 10) (S5 Fig). The results of the MD simulations are summarised in Table 4 provided below. A superimposition of the protein-ligand complexes before and after the simulation has been provided below (Fig 6).

## An overview of the proteins chosen for MD simulation

The proteins that were found to be conserved from the previous analyses were studied in detail. The interaction map of these SARS-CoV-2 proteins from the study by Gordon et al., 2020 reveals targets for drug repurposing [16].

## Nucleoprotein

The nucleoprotein (N-Protein) is a highly charged, multifunctional, basic protein of 422 amino acids which binds to the viral RNA during the virion assembly and leads to formation of the helical nucleocapsid [17]. The N protein and spike protein (S-protein) are encoded by all coronaviruses. The nucleocapsid (N) protein of COVID-19 has nearly 90% amino acid sequence identity with SARS-CoV [18]. However, we observed that the spike protein is not conserved in different variants of SARS-CoV-2 above 90% threshold. The N protein forms complexes with genomic RNA and creates a capsid around the enclosed nucleic acid [17]. It

**Table 4. A table illustrating the mean of various structural parameters for the simulated proteins and protein-ligand complexes.**

| Complex | RMSD (nm) | RMSF (nm) | Radius of Gyration (nm) | SASA (nm^2) | H-bonds |
|---|---|---|---|---|---|
| Free NSP-10 | 0.470361 | 0.20984 | 1.37233 | 66.6657 | - |
| NSP10-Alvimopan | 0.525528 | 0.19471 | 1.38811 | 67.3879 | 0 |
| NSP10-Irbesartan | 0.413957 | 0.17612 | 1.40021 | 70.351 | 1 |
| Free Nucleoprotein | 0.24749 | 0.20316 | 1.45041 | 73.9592 | - |
| Nucleoprotein-Nebivolol | 0.293992 | 0.21579 | 1.46075 | 77.0852 | 3 |
| Nucleoprotein-Bictegravir | 0.342362 | 0.21871 | 1.44189 | 74.0045 | 2 |
| Free 3CL Protein | 0.252472 | 0.16319 | 2.44544 | 232.278 | - |
| 3CL pro-Darifenacin | 0.237117 | 0.16895 | 2.45577 | 237.846 | 1 |
| 3CL pro-Nebivolol | 0.244245 | 0.16176 | 2.44327 | 234.738 | 1 |

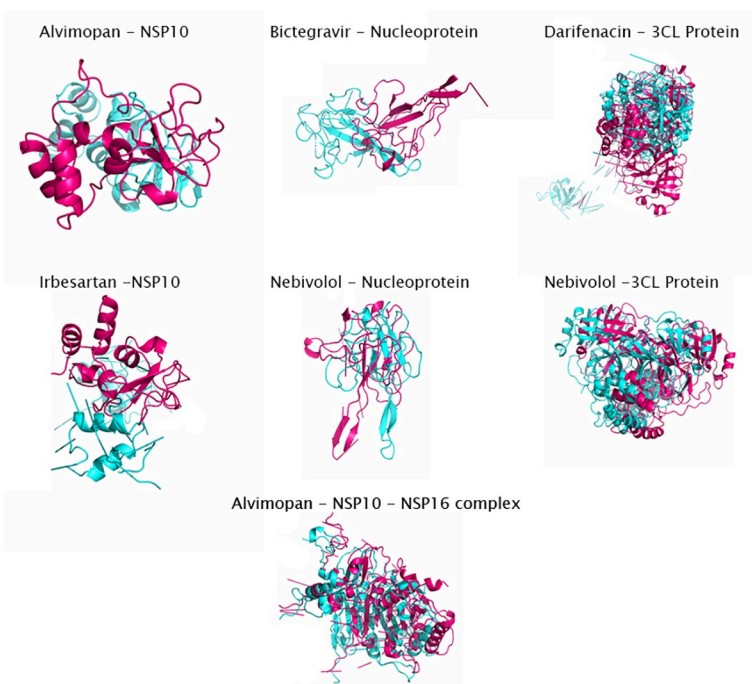

**Fig 6. A superimposition of the protein-ligand complexes before and after the MD simulation.** The protein-ligand complex before the MD simulation is shown in magenta while the complex after the simulation is shown in cyan.

also assists in RNA synthesis and affects the host cell responses such as cell cycle and translation [19]. It plays an important role in virion assembly and enhances the efficiency of the virus transcription and assembly [19]. The interaction map of N-protein reveals that the N-protein interacts with human protein that are responsible for RNA processing and Stress Granule Regulation [16]. This indicates that similar to the N-protein of SARS-CoV, the N-protein of SARS-CoV-2 also plays an important role in suppressing the RNA interference (RNAi) to overcome the host defence. Previous studies have shown that 15 human proteins interact with the N-protein of SARS-CoV-2 [16]. Out of the 15 human proteins interacting with the N-protein, CSNK2B, CSNK2A2 and LARP1 might be plausible drug targets. The drugs chosen for Nucleoprotein were Bictegravir and Nebivolol.

**Root Mean Square Deviation (RMSD).** The Root Mean Square Deviation (RMSD) analysis is an important step towards measuring the stability of the protein-ligand complex. A stable RMSD indicates that the binding of the protein-drug complex does not cause any significant changes in the structure of the protein.

It is evident that the RMSD of the Free Nucleoprotein, Nucleoprotein-Bictegravir, and Nucleoprotein-Nebivolol has remained mostly stable throughout the simulation. The free Nucleoprotein stabilized at around 35 ns and remained stable throughout the simulation. The RMSD of the Nucleoprotein-Nebivolol complex on the other hand stabilized much earlier at around 10 ns and maintained stability throughout except for minor troughs between 20 ns and 40 ns. The RMSD of the Nucleoprotein-Bictegravir complex also stabilized earlier at around 10 ns and remains stabilized except for a small spike at around 70 ns (Fig 7A).

**The radius of gyration (Rg).** The radius of gyration is a key parameter of the Protein-Drug complex that is used to study the folding properties and conformations of the protein-drug complexes. A comparatively high radius of gyration value indicates that a protein molecule is packed loosely while a lower radius of gyration value indicates a protein structure that is

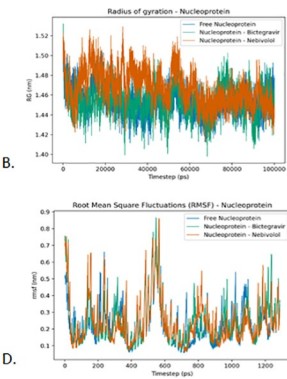

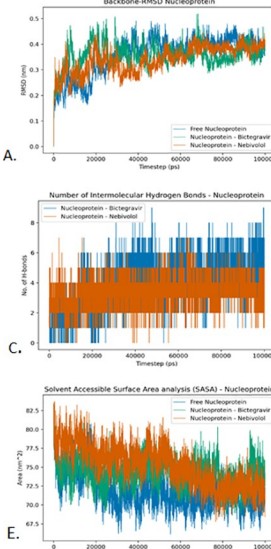

**Fig 7. Analysis of RMSD, radius of gyration, hydrogen bonding, RMSF and SASA of nucleoprotein and drugs Bictegravir and Nebivolol.** A. Root-mean-square deviation of the Cα atoms, B. Radius of gyration (Rg) over the entire simulation, where the ordinate is Rg (nm) and the abscissa is time (ps), C. Total number of H-bond count throughout the simulation, D. RMSF values over the entire simulation, where the ordinate is RMSF (nm) and the abscissa is residue, and E. Solvent accessible surface area (SASA), where the ordinate is SASA (nm$^2$) and the abscissa is time (ps).

more compact. A more compact protein indicates that the drug molecule has not significantly interfered with the folding mechanism of the protein. he radius of gyration of Nucleoprotein-Bictegravir complex and the Nucleoprotein-Nebivolol complex is found to be close to that of the unbound protein. The average Rg value of the unbound Nucleoprotein and Nucleoprotein-Bictegravir complex and Nucleoprotein-Nebivolol complex is found to be 1.454 nm, 1.455 nm, and 1.467 nm respectively. However, this difference in the mean radius of gyration between drugs is not significant as they are well within the standard deviation of the respective complexes. The minor variations in the radius of gyration can be attributed to the conformational changes that the protein-drug complex undergoes (Fig 7B).

**Intermolecular hydrogen bonding.** The number of intermolecular hydrogen bonds is an important parameter that can be used to quantify the binding affinity between the protein and the drug molecule. The presence of a large number of H-bonds between protein and drug molecules signifies a strong binding between the molecules. We observed the maximum number of 9 hydrogen bonds between the protein and drug in the Nucleoprotein-Bictegravir complex and a maximum of 7 in the Nucleoprotein-Nebivolol complex. The average value of intermolecular H-bonds is 4 for Nucleoprotein-Nebivolol complex while 3 for Nucleoprotein-Bictegravir complex. The significant number of hydrogen bonds shows that drug molecules have a high affinity towards the active site of Nucleoprotein (Fig 7C).

**Root Mean Square Fluctuations (RMSF).** Root Mean Square Fluctuations (RMSF) is a vital structural parameter that is used to quantify the flexibility and rigidity of the protein-drug complexes. Since the RMSF measures the deviations of residue from its initial position, it is also highly useful in exploring the conformational flexibility of the protein-drug complexes. In all the proteins, the RMSF at the binding sites was below 0.3 nm. This indicates that the drugs kept close contact with their binding pockets during the MD simulations. In the case of Nucleoprotein, we observed the highest fluctuations between 400–700 atoms stretch. The average RMSF values of Nucleoprotein, Nucleoprotein-Bictegravir complex and Nucleoprotein-

Nebivolol were found to be 0.236 nm, 0.244 nm and 0.243 nm respectively. Further, the RMSF of most residues of the protein is found to be stable below 0.3 nm thereby preserving the flexibility of the protein (Fig 7D).

**Solvent Accessible Surface Area analysis (SASA).** To better understand the solvent Hydrophobic and Hydrophilic behaviour of the protein-drug complexes, solvent accessible surface area analysis (SASA) was performed. These results indicated that all the proteins-ligand complexes are well solvated after the binding of drug molecules. The Solvent Accessible Surface Area analysis revealed that no major differences are observed in the SASA profiles of Nucleoprotein and its protein-drug complexes. The mean SASA values for the free Nucleoprotein, Nucleoprotein-Bictegravir complex and Nucleoprotein-Nebivolol complex were 72.01 $nm^2$, 74.34 $nm^2$, and 75.23 $nm^2$ respectively (Fig 7E).

## 3CLpro

3-chymotrypsin-like cysteine protease (3CLpro) or the NSP5 is also a non-structural protein encoded by ORF1a/1b. The SARS-CoV2 replication process involves a series of proteolytic cleavage of the polypeptide to generate various proteins [7]. The 3CL protease is known to play a critical role at 11 distinct cleavage sites, and is essential for the viral replication [4]. The interaction map of 3CLpro reveals only one human protein-HDAC2, which removes the acetyl groups from lysine residues of core histones [16].HDAC2 plays an essential role in regulating the epigenetic features and gene expression patterns in human cells. All of the above make 3CLpro a suitable target for anti-coronavirus drugs. The drugs that were docked with a higher score the 3Clpro were Darifenacin and Nebivolol.

**Root Mean Square Deviation (RMSD).** From the RMSD plot of 3CLpro, we can see that the free form of the protein and the 3CLpro-Darifenacin complex stabilizes at around 25 ns. While the free protein remains stabilized till the end, the 3CLpro-Darifenacin complex has a few minor instabilities between 70 ns and 80 ns. On the other hand, the 3CLpro-Nebivolol complex stabilizes earlier at around 10 ns and stays stabilized throughout the simulation (Fig 8A).

**The radius of gyration (Rg).** The compactness of the protein is found to be unaffected by the binding of the drugs as they have a similar radius of gyration. The average Radius of gyration value of Unbound 3CLpro Protein, 3CLPro-Darifenacin, and 3CLPro-Nebivolol complex is found to be 2.449 nm, 2.467 nm, and 2.467 nm respectively. The differences in the radius of gyration are well within the standard deviation of the respective proteins. We also observe a gradual decrease in the Radius of gyration value of the protein-ligand complexes. This indicates that the secondary structure of the protein is not significantly affected by the binding of the drugs (Fig 8B).

**Intermolecular hydrogen bonding.** The maximum number of intermolecular hydrogen bonds in the 3CLpro-Darifenacin complex and the 3CLpro-Nebivolol complex is found to be 4 and 7 respectively. The average number of intermolecular H-bonds for both 3CLpro-Nebivolol complex and 3CLpro-Darifenacin complex was found to be 1. Unlike the 3CLpro-Darifenacin complex where Hydrogen bonds can be observed since the start of the simulation, the hydrogen bonds in 3CLpro-Nebivolol complex start appearing only after 13ns (Fig 8C).

**Root Mean Square Fluctuations (RMSF).** In the case of 3CLpro, we observe high fluctuations throughout the protein chain in both free protein and protein-drug complexes. No major differences are observed in the RMSF profiles of the free protein and protein-drug complexes. The average RMSF values of 3CLpro, 3CLpro-Darifenacin complex and 3CLpro-Nebivolol were found to be 0.158 nm, 0.180 nm, and 0.156 nm respectively (Fig 8D). These RMSF

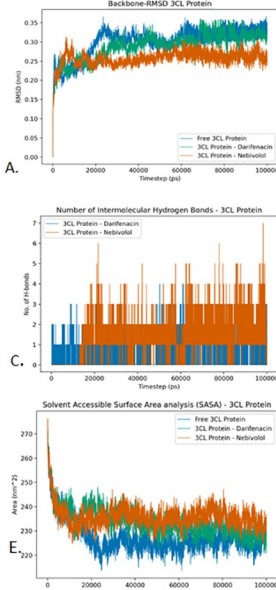
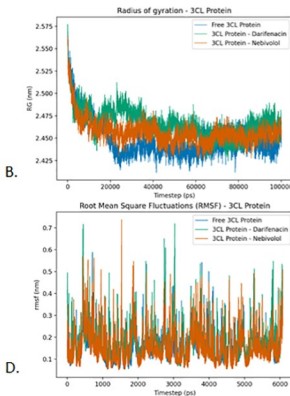

**Fig 8. Analysis of RMSD, radius of gyration, hydrogen bonding, RMSF and SASA of 3CLpro protein and drugs Darifenacin and Nebivolol.** A. Root-mean-square deviation of the Cα atoms, B. Radius of gyration (Rg) over the entire simulation, where the ordinate is Rg (nm) and the abscissa is time (ps), C. Total number of H-bond count throughout the simulation, D. RMSF values over the entire simulation, where the ordinate is RMSF (nm) and the abscissa is residue, and E. Solvent accessible surface area (SASA), where the ordinate is SASA (nm$^2$) and the abscissa is time (ps).

values indicate that the binding of Darifenacin and Nebivolol preserve the flexibility of the protein.

**Solvent Accessible Surface Area analysis (SASA).**   The average SASA values of 3CLpro, 3CLpro-Darifenacin complex and 3CLpro-Nebivolol complex are found to be 227.64 nm$^2$, 233.85 nm$^2$, and 235.92 nm$^2$, respectively (Fig 8E).

## NSP10

NSP10 is one of the 16 non-structural proteins (NSP1–16) encoded by ORF1a/1b that comprise the RNA-synthesizing machinery of SARS-CoV2. The NSP10 subunit contains two zinc fingers and is known to interact with the NSP14 and NSP16 subunits to increase their 3′-5′ exoribonuclease and 2′-O-methyltransferase activities respectively [20]. Existing literature suggests that the NSP10/14 interaction is crucial for the viral replication process as mutations in NSP10 that abolished the interaction are known to have yielded replication-negative virus [20]. The network map for NSP10 reveals that the protein interacts with several proteins responsible for endomembrane compartments and vesicle trafficking pathways [16]. Among these human-proteins are the AP2 (AP2A2 and AP2M1) proteins that are associated with cla-thrin-mediated endocytosis [16]. Interaction of NSP10 with these human-proteins are hypothesized to modify endomembrane compartments to favor coronavirus replication [16]. Among the FDA approved drugs, that were screened for NSP10, Alvimopan and Irbesartan had a higher docking score and were subjected to further MD analysis. Since NSP10 is also known to make complex with NSP16, we did the screening of the drugs for the NSP 10–16 complex. This was further subjected to MD analysis to look for the stability of the drug binding to the complex. While the binding of Alvimopan with the complex was stable, the NSP-Irbesartan

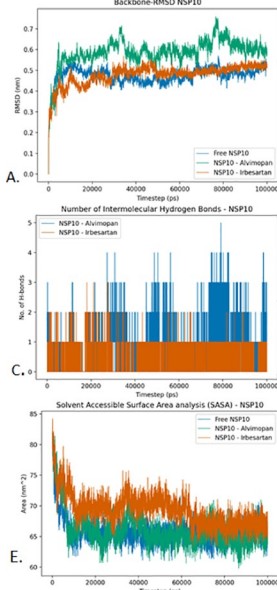
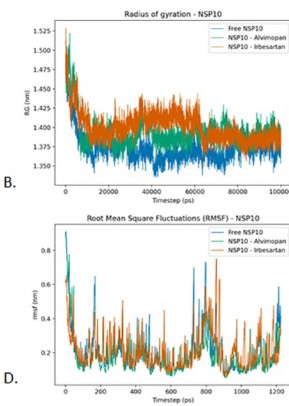

**Fig 9. Analysis of RMSD, radius of gyration, hydrogen bonding, RMSF and SASA of NSP10 protein and drugs Alvimopan and Irbesartan.** A. Root-mean-square deviation of the Cα atoms, B. Radius of gyration (Rg) over the entire simulation, where the ordinate is Rg (nm) and the abscissa is time (ps), C. Total number of H-bond count throughout the simulation, D. RMSF values over the entire simulation, where the ordinate is RMSF (nm) and the abscissa is residue, and E. Solvent accessible surface area (SASA), where the ordinate is SASA (nm$^2$) and the abscissa is time (ps).

complex was not stable as was evident from the MD analysis. Various parameters used in the MD analysis for the individual proteins are mentioned in Table 4.

**Root Mean Square Deviation (RMSD).** *NSP10 protein.* Fig 9A reveals that the RMSD of the Free NSP10 protein, NSP10-Alvimopan, and NSP10-Irbesartan complexes are stabilized. The RMSD of the Free NSP10 protein stabilizes at around 7 ns and maintains stability until the end. The NSP10-Alvimopan complex attains stability at around 10 ns and remains stable throughout the simulation barring small spikes at around 30 ns and 80 ns. The NSP10-Irbesartan complex, on the other hand, reaches stability comparatively later at around 15 ns and remains stabilized throughout. These results indicate that the drugs did not significantly influence the structural stability of the NSP10 protein. In particular, the NSP10-Irbesartan complex has an average RMSD that is very close to the RMSD of the drug-free form of NSP10.

*NSP10-NSP16 complex.* Looking at Fig 10A we can see that the NSP10-NSP16 complex has stabilized both in the free form and docked form. The Free NSP10-NSP16 complex stabilizes at around 10 ns and stays stabilized throughout the simulation. The NSP10-NSP16 Alvimopan complex stabilizes a little later at around 40 ns and stays stabilized throughout the simulation.

**The radius of gyration (Rg).** *NSP10 protein.* The mean radius of gyration for the Free-NSP10, NSP10-Alvimopan complex, and NSP10-Irbesartan complex is found to be 1.373, 1.393, and 1.401 respectively. Although the mean radius of gyration indicates that the NSP10-Irbesartan and NSP10-Alvimopan complexes are not as compact as the Free-NSP10 complex. The radius of gyration plot (Fig 9B) reveals that (after 60 ns) the final conformations of the Free-NSP10 and NSP10-Alvimopan complex have a very similar radius of gyrations. This indicates that the binding of Alvimopan has not affected the folding of the protein. The binding of Irbesartan on the other hand slightly affects the folding of the protein (Fig 9B).

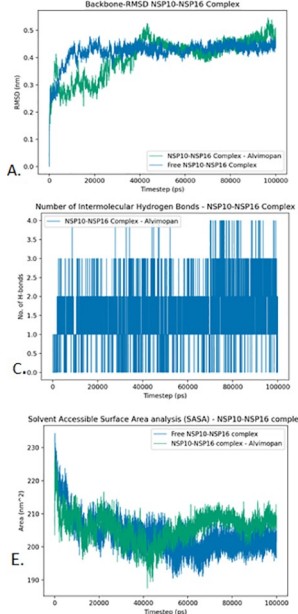

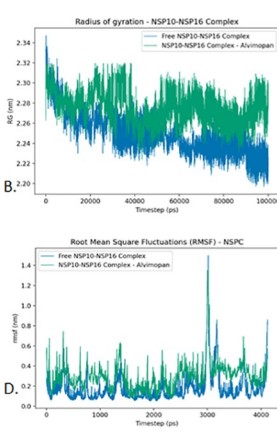

**Fig 10. Analysis of RMSD, radius of gyration, hydrogen bonding, RMSF and SASA of NSP10-16 complex and drug Alvimopan.** A. Root-mean-square deviation of the Cα atoms, B. Radius of gyration (Rg) over the entire simulation, where the ordinate is Rg (nm) and the abscissa is time (ps), C. Total number of H-bond count throughout the simulation, D. RMSF values over the entire simulation, where the ordinate is RMSF (nm) and the abscissa is residue, and E. Solvent accessible surface area (SASA), where the ordinate is SASA ($nm^2$) and the abscissa is time (ps).

*NSP10-NSP16 complex.* The Mean radius of gyration of Free NSP10-NSP16 complex and the Alvimopan docked NSP10-NSP16 complex was found to be 2.250 and 2.276 respectively. This difference is well within the standard deviation of the respective complexes. However, the radius of gyration plot reveals that the final structure of the Alvimopan docked NSP10-NSP16 complex is less compact than the free NSP10-NSP16 complex (Fig 10B).

**Intermolecular hydrogen bonding.** *NSP10.* In the case of NSP10, NSP10-Alvimopan and NSP10-Irbesartan complexes have a maximum of 5 and 3 Hydrogen bonds respectively. In both the cases, the average number of intermolecular hydrogen bonds between protein and drug is found to be 1. The plots indicate that the Drug-protein affinity is higher in the case of Alvimopan than in the case of Irbesartan (Fig 9C).

*NSP10-NSP16 complex.* The Alvimopan docked NSP10-NSP16 complex was found to have a maximum of 4 Hydrogen bonds in the simulation. The mean number of hydrogen bonds between the Alvimopan and NSP10-NSP16 complex is found to be around 2. These results suggest a considerable affinity between the drug and the Protein complex (Fig 10C).

**Root Mean Square Fluctuations (RMSF).** *NSP10.* The RMSF profile of NSP10 and its complexes reveal that the protein has high fluctuations in the 0 to 100 stretch and in the 600 to 1000 stretch. The overall RMSF profile of free NSP10 is found to be similar to that of the drug-complexes. The average RMSF of free NSP10, NSP10-Alvimopan and NSP10-Irbesartan was found to be 0.210, 0.187, and 0.215 respectively. This indicates that there might be a slight loss of flexibility from the binding of the drug molecules (Fig 9D).

*NSP10-NSP16 complex.* The RMSF profile of NSP10-NSP16 complex and its Alvimopan docked form reveal that the docked form of the protein complex has higher fluctuations compared to the free form. The Alvimopan docked NSP10-NSP16 complex has considerably higher RMSF in the 0 to 100 residues stretch and in the 600 to 1000 residue stretch. The average RMSF of free NSP10-NSP16 complex and the Alvimopan NSP10-NSP16 complex was

found to be 0.179 and 0.293 respectively. This indicates that there might be a slight loss of flexibility from the binding of the drug molecules (Fig 10D).

**Solvent Accessible Surface Area analysis (SASA).** In the case of NSP10, the Free NSP10 Protein, NSP10-Alvimopan complex, and NSP10-Irbesartan complex are found to have average SASA values of 65.90 nm$^2$, 66.14 nm$^2$, and 69.33 nm$^2$ respectively (Fig 9E). In the case of NSP10-NSP16 complex, the Free NSP10-NSP16 complex and Alvimopan docked NSP10-NSP16 complex was found to have average SASA values of 203.57 nm$^2$ and 205.88 nm$^2$ (Fig 10E). In all these cases, the drug docked complexes were found to be better solvated compared to the free versions of the proteins. This result can be attributed to the larger radius of gyration of the drug docked complexes. Therefore, no major differences are observed in the SASA profiles of these complexes.

## Side-effects of the drugs chosen for targeting

The drugs selected for repurposing are Alvimopan, Nebivolol, Darifenacin, Irbesartan and Bictegravir. 3CLpro is targeted by Darifenacin and Nebivolol, NSP10 is targeted by Irbesartan and Alvimopan and Nucleoprotein is targeted by Nebivolol and Bictegravir. Alvimopan, which is a mu-opioid receptor antagonist, is used for accelerating upper and lower gastrointestinal tract recovery after a bowel resection [21]. Nebivolol is a beta blocker that is used to treat hypertension and heart failure [22]. Bictegravir is an integrase inhibitor class viral drug that is used to treat HIV and other retroviral diseases [23]. Irbesartan is an angiotensin receptor blockers used in the treatment of hypertension and also to protect the kidneys from damage due to diabetes [24].[43]. Darifenacin is a medication to treat urinary incontinence [25]. It interacts with the M3 muscarinic acetylcholine receptors, which mediate bladder muscle contractions [26]. The side effects of these drugs were analysed from the SIDER database of drugs and side effects (http://sideeffects.embl.de/about/). This revealed that the major side effects of these FDA approved drugs are Headache, Dizziness, Diarrhoea and Constipation. In addition to these, back pain and dry mouth were also observed in the case of Darifenacin.

Alvimopan which has a high binding affinity (Ki = 0.4 nM) and a low dissociation rate (half-life = 30–44 min) has a low bioavailability of 6%. It reaches a maximum plasma concentration within two hours of administration [27] Nebivolol has the oral bioavailability of 12% with the half-life of nearly 10 hours in Extensive Metabolizers (EMs) and the oral bioavailability of 96% with the half-life of nearly 32 hours in Poor Metabolizers (PMs) [28]. Darifenacin has the absolute bioavailability of 15.4% and 18.6% for 7.5 mg and 15 mg prolonged-release tablets respectively [29]. Irbesartan which is administered orally has an average bioavailability ranging from 60% to 80% [30]. Bictegravir has a bioavailability of greater than 70% and a median plasma half-life of 18 hours after one dosage [31].

## Conclusion

The mutation events in the viral genome contribute to structural changes in the proteins thereby making it a difficult therapeutic target. This is one of the essential parameters which needs to be reconsidered in the drug-development process for a successful and effective design of therapeutics. By using our stable gene approach we suggest Darifenacin, Nebivolol, Bictegravir, Alvimopan and Irbesartan as potential drugs for the clinical trials against SARS-CoV-2. Further, a BLAST search of human proteins with the selected SARS-CoV-2 proteins indicates that there are no human proteins that are similar to shortlisted viral proteins minimising the off target binding of the drugs.

The SARS-CoV-2 pandemic was declared a Public Health Emergency of International Concern (PHEIC) by the World Health Organisation (WHO) on 30th January, 2020. Since then

there have been several studies regarding drug designing and appropriate pre-clinical and clinical trials for drugs and vaccines. This particular study finds its significance in utilising the conserved genes as stable targets for drug designing which gives a greater confidence while testing the drugs in the clinical trials. The drugs Darifenacin, Nebivolol, Bictegravir, Alvimopan and Irbesartan targeted the stable genes 3CLpro, Nucleoprotein and NSP10 and were shown to stabilize the Drug-Protein complex in MD simulations. We also find the mutation frequency across the viral genome, the conserved genes and the population level variant genes which would greatly benefit the designing of vaccines and cure for SARS-CoV-2. Our haplotype network gives an impression of seven different viral strains spread across the globe with different frequencies and phylogenetic tree raises concerns about its origin. The drugs reported in this paper can be further analysed and used as an antiviral drug against SARS-CoV-2 upon further downstream analysis and appropriate clinical trials.

## Materials and methods

The complete high throughput FASTA file for 358 nCOV2 viral genomes were downloaded on 15[th] June 2020 from GISAID (Global Initiative on Sharing All Influenza Data; https://www.gisaid.org/) with acknowledgment (S1 File). These primary group of viral genomes represent the ancestral class for the further evolving strains. While the number of sequences has increased drastically over time, this set of 358 genomes represent the genome sequenced from diverse regions. Sequence and annotation of the reference genome of SARS-CoV-2 (NC_045512.2) was downloaded from GenBank and GISAID.

### Haplotype network

We used DnaSP v6.12.03 to define sequence sets and generate multi-sequence aligned haplotype data in nexus file format [32]. In the nexus data file the trait segment was included for visualisation and drawing of haplotype networks based on the haplotypes generated by the DnaSP. We then further used PopART v1.7 to draw the haplotype network based on the haplotype by DnaSP [33].

### Conserved gene and population level variants

The 358 sequences from humans were aligned using online MAFFT's closely related viral genome alignment tool [34] with the reference sequence NC_045512.2. The FFT-NS-fragment method was used for alignment with the parameters—reorder—adjustdirection—keeplength —mapout—anysymbol. Default gap penalty of 1.53 and offset value of 0.0 was used. The number of mutations were counted for each nucleotide position using NC_045512.2 as reference. The ambiguous bases, Ns and gaps were not treated as mutations.

The sliding frame method was used to identify conserved genes across the given alignment. The programs used for this analysis are publicly shared on GitHub (https://github.com/DevangLiya/CRAM). A master sequence was produced consisting of 1 for the nucleotide position that is conserved across all the genomes and 0 for the nucleotide position that is mutated in at least one genone. A frame of size 100 was moved across the entire length of this master sequence and each instance of frame was given a score between 0 and 100 based on the number of 1s in that instance of frame. We call this "conservation score". Starting position of every frame with the conservation score between the given thresholds is reported. The nucleotide sequence corresponding to these conserved frames were reconstructed by adding 100 nucleotides (equal to the frame length) to the reported positions and the sequence was then BLASTed to get the corresponding genes [35]. A few more nucleotides were added on the both ends of the sequence when BLAST did not yield any satisfactory match.

The dataset of 358 sequences was divided into the eight population level datasets consisting of China, Japan, Asia (India, Singapore, Cambodia, Nepal, Vietnam, Taiwan, Hong Kong, Thailand, South Korea), Europe (France, Finland, Netherlands, Czech Republic, Switzerland, Italy, Portugal, Germany, Luxembourg, Sweden, Belgium), UK (England, Wales, Ireland), North America (USA and Canada), Oceania (Australia and New Zealand), and Rest of America (Mexico, Chile, Brazil). These sequences were then aligned and visualized in MEGA to identify population level mutations.

## Protein structure modelling

The 3D structure of the respective proteins were modelled with the best reported NMR structures as their template for homology modelling. The crystal structures of the protein complexes were availed as the template for modelling individual 3D structures. The 3D structure models for the proteins screened were modelled by comparative protein modelling methods using the SWISS-MODEL server (http://swissmodel.expasy.org) [36]. The structure-based alignment obtained were used and SWISS-MODEL was used in the optimized mode to minimize energy. Models are made according to the target template alignment and the per-residue and the global model quality was assessed using the QMEAN and Global Model Quality Estimate (GMQE) scoring functions. The GMQE score gives an estimate of accuracy of tertiary structure of the protein models. The QMEAN on the other hand gives an impression of the quality of the submitted model based on its physicochemical properties and then generates a value referring to the overall quality of the structure.

## Validation of models

RAMPAGE was used for the Ramachandran Plot analysis and for the verification of 3D structures. It provides the number of residues in the favored, allowed, and outlier region [37]. If a good proportion of residues lie in the favored and allowed region, then the model is predicted to be good. The quality of the models were also assessed using ProSA, PROCHECK and Verify 3D [38–40]. Both PROCHECK and RAMPAGE analyze the stereo chemical quality of the submitted models based on its phi/psi angle arrangement and then generates Ramachandran plots which highlights the percentage of residues in the favored, allowed or in outlier regions. If a greater proportion of the residues lie in the favored and allowed region then the model is considered to be good. ProSA on the other hand does a comparative analysis by calculating the potential energy of the protein models and comparing them to the experimental structures deposited in the PDB. The Z-Scores obtained from each model suggest that the structures are comparable to the NMR structures of similar size. Verify3D evaluates the local quality of the protein model on the basis of structure-sequence compatibility to generate a compatibility value for each residue of the protein. A model with 80% of their residues with a 3D-1D score equal to or higher than 0.2 is considered to be a high quality structure.

## Virtual screening and molecular docking

A comparison with other docking and screening platforms such as AutoDock$_4$, AutoDock$_{4Zn}$, AutoDock Vina, Quick Vina 2, LeDock, and UCSF DOCK6, shows that PLANTS (Protein-Ligand ANT System) has most accurate posing algorithms for the protein-ligand docking [41]. In order to perform a structure-based virtual screening e-LEA3D, (http://chemoinfo.ipmc.cnrs.fr/) which uses PLANTS algorithm was used. In order to find the binding site around a residue metaPocket 2.0 software was used [42]. The virtual screening was done on the basis of docking with the list of FDA approved drugs. The docking score provided by the e-LEA3D

was used to screen the drugs for further analysis. From the docking scores the two drugs which had a higher docking score were chosen for the MD analysis.

## Molecular dynamic simulations

The unbound proteins and Protein-drug complexes were subjected to MD simulation for 100 ns to mimic the physiological state of protein molecules. The simulation was performed with GROMACS 2019 (M.J. Abraham) utilizing the GROMOS96 43a1 force field parameters [43]. The topologies of the drug molecules were modelled using the PRODRG web server [44]. The system was made electrostatically neutral by adding counter ions and the complexes were solvated within 10 SPC/E water cube [43, 45]

The whole system was then energy minimized in multiple steps using the steepest descent method. The temperature of the entire system was raised up to 300 K for a time scale of 100 ps. Two different phases of equilibration were performed-first with constant pressure and temperature (NPT) and the other with steady volume and temperature (NVT) [46]. The trajectory file of simulated system was then used for calculation of various structural parameters like the Root Mean Square Deviation (RMSD), Root Mean Square Fluctuations (RMSF), Radius of Gyration (Rg), Intermolecular Hydrogen Bonding (H-bonding) and Solvent-Accessible Surface Area (SASA) to understand the structural behaviour of the protein-drug complexes [47].

## Supporting information

**S1 Fig. Validation of the predicted model of 3CLpro by ProSa and Verify3D.** A. Validation of structure by ProSa, which compares the predicted model of 3CLpro (black dot), with a non-redundant set of crystallographic structures (light blue dots) and NMR structures (dark blue dots) and provides a Z-score. B. Validation of structure by Verify3D, which highlights the 3D-1D score for every atom of the predicted model. 94.28% of the residues in the 3CLpro model had a compatibility score of 0.2 or higher, which indicates a high quality structure.
(TIF)

**S2 Fig. Validation of the in silico predicted model of NSP10 by ProSa and Verify3D.** A. Validation of structure by ProSa, which compares the predicted model of 3CLpro (black dot), with a non-redundant set of crystallographic structures (light blue dots) and NMR structures (dark blue dots) and provides a Z-score B. Validation of structure by Verify3D, which highlights the 3D-1D score for every atom of the predicted model. 82.44% of the residues in the NSP10 model had a compatibility score of 0.2 or higher, which indicates a high-quality structure.
(TIF)

**S3 Fig. Validation of the in silico predicted model of nucleoprotein by ProSa and Verify3D.** A. Validation of structure by ProSa, which compares the predicted model of 3CLpro (black dot), with a non-redundant set of crystallographic structures (light blue dots) and NMR structures (dark blue dots) and provides a Z-score.B. Validation of structure by Verify3D, which highlights the 3D-1D score for every atom of the predicted model. 94.49% of the residues in the nucleoprotein model had a compatibility score of 0.2 or higher, which indicates a high-quality structure.
(TIF)

**S4 Fig. Validation of the in silico predicted model of PLpro by ProSa and Verify3D.** A. Validation of structure by ProSa, which compares the predicted model of 3CLpro (black dot), with a non-redundant set of crystallographic structures (light blue dots) and NMR structures (dark blue dots) and provides a Z-score.B. Validation of structure by Verify3D, which

highlights the 3D-1D score for every atom of the predicted model. 95.85% of the residues in the PLpro model had a compatibility score of 0.2 or higher, which indicates a high-quality structure.
(TIF)

**S5 Fig. Drug-protein interaction after docking.** NSP 10–16 complex interaction with Alvimopan. Drugs are in orange while the NSP16 is highlighted in purple and NSP10 is labelled in blue. The residues interacting with the drugs are highlighted in light blue.
(TIF)

**S1 File.**
(XLS)

## Acknowledgments

We would like to thank GISAID (Acknowledgments table in the S1 File) for the database of SARS-CoV-2 genome sequences. The authors would like to thank Shubham Kumar Sinha and Mirudula Elanchezhian for their help with global conservation analysis.

## Author Contributions

**Conceptualization:** Arpit Kumar Pradhan, Ashwin Kumar Jainarayanan.

**Data curation:** Nithishwer Mouroug Anand, Devang Haresh Liya, Arpit Kumar Pradhan, Ashwin Kumar Jainarayanan.

**Formal analysis:** Nithishwer Mouroug Anand, Devang Haresh Liya, Arpit Kumar Pradhan, Ashwin Kumar Jainarayanan.

**Investigation:** Nithishwer Mouroug Anand, Devang Haresh Liya, Arpit Kumar Pradhan, Ashwin Kumar Jainarayanan.

**Methodology:** Arpit Kumar Pradhan, Ashwin Kumar Jainarayanan.

**Resources:** Arpit Kumar Pradhan, Ashwin Kumar Jainarayanan.

**Software:** Devang Haresh Liya.

**Supervision:** Arpit Kumar Pradhan, Ashwin Kumar Jainarayanan.

**Validation:** Nithishwer Mouroug Anand, Devang Haresh Liya, Arpit Kumar Pradhan, Nitish Tayal, Sainitin Donakonda, Ashwin Kumar Jainarayanan.

**Visualization:** Nithishwer Mouroug Anand, Devang Haresh Liya, Nitish Tayal, Abhinav Bansal.

**Writing – original draft:** Nithishwer Mouroug Anand, Devang Haresh Liya, Arpit Kumar Pradhan, Nitish Tayal, Ashwin Kumar Jainarayanan.

**Writing – review & editing:** Nithishwer Mouroug Anand, Devang Haresh Liya, Arpit Kumar Pradhan, Ashwin Kumar Jainarayanan.

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
