## [Decision Letter · Decision Letter 0]

11 Dec 2020

PONE-D-20-34651

A comprehensive SARS-CoV-2 genomic analysis identifies potential targets for drug repurposing

PLOS ONE

Dear Dr. Pradhan,

Thank you for submitting your manuscript to PLOS ONE. After careful consideration, we feel that it has merit but does not fully meet PLOS ONE’s publication criteria as it currently stands. Therefore, we invite you to submit a revised version of the manuscript that addresses the points raised during the review process.

While one reviewer found it acceptable at the current form, the others found some concerning issues or misleading/vague/improvable sentences.

As suggested, please describe the docking results with hsp10-16 dimer, and refer to other phylogenetic studies instead of redoing them in this. 

In addition to what the reviewers have asked, please review the manuscript yourself and correct or fix anything that you think would improve the readers understanding with minimum navigation within the manuscript. You can add these as additional edits.

A rebuttal letter that responds to each point raised by the academic editor and reviewer(s). You should upload this letter as a separate file labeled 'Response to Reviewers'. Please refer to the line(s) or paragraph(s) in the final revised document. For most of the sentence edits it would be useful if you can provide the modified portion in the answer as:"We corrected/modified the sentence, which now reads: xxx yyy zzz...."A marked-up copy of your manuscript that highlights changes made to the original version. You should upload this as a separate file labeled 'Revised Manuscript with Track Changes'.An unmarked version of your revised paper without tracked changes. You should upload this as a separate file labeled 'Manuscript'.

We look forward to receiving your revised manuscript.

Kind regards,

Malaya Kumar Sahoo, Ph.D

Academic Editor

PLOS ONE

Journal Requirements:

2. Thank you for submitting the above manuscript to PLOS ONE. During our internal evaluation of the manuscript, we found significant text overlap between your submission and the following previously published works:

https://www.biorxiv.org/content/10.1101/2020.10.30.362764v1

https://www.pnas.org/content/117/17/9241.long

Please revise the manuscript to rephrase the duplicated text within the manuscript and Figure captions, cite your sources, and provide details as to how the current manuscript advances on previous work. Please note that further consideration is dependent on the submission of a manuscript that addresses these concerns about the overlap in text with published work.

3. During our initial internal evaluation of your submission, we noticed several typos in your manuscript. Please note that PLOS ONE does not provide copyediting or proofs of accepted manuscripts. We therefore recommend that you carefully review your manuscript and correct any errors at this time.

4. Thank you for stating the following in the Financial Disclosure section:

'The author(s) received no specific funding for this work.'

We note that one or more of the authors are employed by a commercial company: The Unique Tutorials

6. Please remove your figures from within your manuscript file, leaving only the individual TIFF/EPS image files, uploaded separately.  These will be automatically included in the reviewers’ PDF.

Reviewers' comments:

Reviewer's Responses to Questions

**Comments to the Author**

1. Is the manuscript technically sound, and do the data support the conclusions?

Reviewer #1: Yes

Reviewer #2: Partly

Reviewer #3: Partly

2. Has the statistical analysis been performed appropriately and rigorously? 

Reviewer #1: Yes

Reviewer #2: I Don't Know

Reviewer #3: No

3. Have the authors made all data underlying the findings in their manuscript fully available?

Reviewer #1: Yes

Reviewer #2: No

Reviewer #3: Yes

4. Is the manuscript presented in an intelligible fashion and written in standard English?

Reviewer #1: Yes

Reviewer #2: Yes

Reviewer #3: No

5. Review Comments to the Author

Reviewer #1: In this Manuscript Anand et al. provide the important findings such as identification of conserved genes and the drugs candidates to target the conserved genes. This approach of identifying stable genes and targeting stable genes with drugs would provide a better therapeutic approach and possible hope in the successive clinical trials.

The manuscript was well written, the experiments well designed, and the conclusions are appropriate. I believe that the findings of the manuscript are of sufficient novelty and breadth to merit publication in PLOS ONE.

Below are suggestions for minor revision of the manuscript:

1. Sentences are too long in the abstract section, I would like to recommend split the sentences in abstract.

One of the recent studies suggests the existence of three central variants of SARS-CoV-2 distinguished by amino acid changes 13. We urgently need therapeutic options to combat this virus infection.

2. The sentences above are important for the introduction part but it would sound better, if the authors can bring some coherence.

3. “This approach of targeting the stable genes for drug discovery would provide a better therapeutic approach and confidence in the successive clinical trials”.

Please check the bioavailability of the identified drugs (in plasma and organ tissues) whether the drug candidates has possible therapeutic efficacy at MTD drug doses. However I have no major concerns with this paper in its current form.

Reviewer #2: In the presented manuscript, Anand et al. authors performed a phylogenetic analysis of SASR-COV-2 variants to determine the most conserved region of the virus and performed a virtual screening against determined proteins FDA-approved drugs. Further, they performed molecular docking and determined the drugs Darifenacin, Nebivolol, Bictegravir, Alvimopan, and Irbesartan as the most promising candidates.

In the first part of the paper, the phylogenetic analyses are unnecessary and misleading. There is a number of high-quality publications (e.g. doi.org/10.1101/2020.04.17.046086 ., DOI: 10.1002/jmv.25902, DOI: 10.1002/jmv.25699) and publicly available resource for phylogenetic analysis of SARS-COV-2 variants https://nextstrain.org/. Therefore, to avoid controversial and unsupported conclusions of coronavirus's potential origin and transmission roads, I recommend focusing on virtual screening and docking of the FDA-approved drugs, presented in the second part of the manuscript.

With the modifications, the paper could be suitable for publication

Reviewer #3: This paper presents a very standard approach to both the phylogenetic analysis of SARS-CoV2 and the homology modeling building and virtual ligand screening. There are a number of issues that preclude publication in the following form:

1. The paper is disjoint. What insights does the phylogenetic analysis provide with respect to the drug repurposing aspect of the algorithm?

2.There are numerous technical and conceptual problems with the ligand screening/homology modeling. 25 ns or so simulations are too short to see significant structural changes. How were the the particular drugs chosen? What is the point of doing a Lipinski analysis on FDA approved drugs?

3. The paper targets nsp10. But nsp10 forms a complex with nsp16 which is likely to be a biologically relevant conformation. How does the dimer affect the putative ligand binding with nsp10? Why not simulate the dimer since there is a crystal structure of the nsp10/nsp16 complex?

4. What is the false positive rate of the docking approach that is used? How confident are the binding predictions?

5. The authors need to perform experimental validation that their predictions are correct.

Overall, this paper represents an incremental contribution to a very important and difficult problem.

6. PLOS authors have the option to publish the peer review history of their article (what does this mean?). If published, this will include your full peer review and any attached files.

Reviewer #1: **Yes: **Raja Sab Kalluru

Reviewer #2: No

Reviewer #3: No

---

## [Author Response · Author response to Decision Letter 0]

31 Jan 2021

Dear Dr. Malaya Kumar Sahoo,

Academic Editor

PLOS ONE

We thank you, and the reviewers, for the detailed review, and the comments on our manuscript [PONE-D-20-34651] - [EMID:e297d3cb56d8414a] entitled “A Comprehensive SARS-CoV-2 Genomic Analysis Identifies Potential Targets for Drug Repurposing”. We have gone through all the comments carefully, and have addressed all the points that were raised by the reviewers in detail. We have modified the paper in response to the remarks, and hope that this revised version of the manuscript is suitable for publication in the journal.

We have appended a detailed response in the following pages.

Thank you.

Best wishes,

Response to Editor’s comments:

Response to comment 1: 

We have redrafted the entire manuscript in accordance to the PLOS ONE’s style guidelines. 

2. Thank you for submitting the above manuscript to PLOS ONE. During our internal evaluation of the manuscript, we found significant text overlap between your submission and the following previously published works:

https://www.biorxiv.org/content/10.1101/2020.10.30.362764v1

https://www.pnas.org/content/117/17/9241.long

Please revise the manuscript to rephrase the duplicated text within the manuscript and Figure captions, cite your sources, and provide details as to how the current manuscript advances on previous work. Please note that further consideration is dependent on the submission of a manuscript that addresses these concerns about the overlap in text with published work.

Response to comment 2 :

We have made sure that there is no duplication of contents from any previously published works. 

3. During our initial internal evaluation of your submission, we noticed several typos in your manuscript. Please note that PLOS ONE does not provide copyediting or proofs of accepted manuscripts. We therefore recommend that you carefully review your manuscript and correct any errors at this time.

Response to comment 3:

We have checked the manuscript extensively to make sure that there are no typos. 

4. Thank you for stating the following in the Financial Disclosure section:

'The author(s) received no specific funding for this work.'

We note that one or more of the authors are employed by a commercial company: The Unique Tutorials

Response to comment 4a :

We apologize for the misunderstanding, we have now corrected the affiliation of the respective author from a commercial company to an academic institute.

Response to comment 4b:

All the authors have declared no conflict of interest. Since the affiliation of the author in question has been corrected, the above concern has been resolved. 

Response to comment 4c:

Funding: The authors have not received any funding for this particular work. This research has been carried out of mere scientific interest. 

Conflict of interest: All the authors have declared no conflict of interest. 

These are now stated in the manuscript. 

Response to comment 5:

The corresponding authors have now provided the ORCiD id for the corresponding authors during the submission.

6. Please remove your figures from within your manuscript file, leaving only the individual TIFF/EPS image files, uploaded separately. These will be automatically included in the reviewers’ PDF.

Response to comment 6:

We have now removed the images from the manuscript and made sure that the figures are uploaded separately in the TIFF/EPS image format. 

Response to Reviewers’ comments:

Reviewer #1: In this Manuscript Anand et al. provide the important findings such as identification of conserved genes and the drugs candidates to target the conserved genes. This approach of identifying stable genes and targeting stable genes with drugs would provide a better therapeutic approach and possible hope in the successive clinical trials.

The manuscript was well written, the experiments well designed, and the conclusions are appropriate. I believe that the findings of the manuscript are of sufficient novelty and breadth to merit publication in PLOS ONE.

Below are suggestions for minor revision of the manuscript:

1. Sentences are too long in the abstract section, I would like to recommend split the sentences in abstract.

Response 1:

The sentences in the abstract have been rephrased.

One of the recent studies suggests the existence of three central variants of SARS-CoV-2 distinguished by amino acid changes 13. We urgently need therapeutic options to combat this virus infection.

2. The sentences above are important for the introduction part but it would sound better, if the authors can bring some coherence.

Response 2:

We have rewritten the introduction to address this. Please refer to the track change version to infer the exact changes.

3. “This approach of targeting the stable genes for drug discovery would provide a better therapeutic approach and confidence in the successive clinical trials”.

Please check the bioavailability of the identified drugs (in plasma and organ tissues) whether the drug candidates has possible therapeutic efficacy at MTD drug doses. 

Response 3:

We have introduced a new paragraph under the headline of “Side-Effects of the drugs chosen for targeting” in the results and discussion section providing detailed information about the bioavailability and MTD dosage for all the repurposed drugs suggested in the manuscript. The references for the respective drugs are added in the bibliography section.

However I have no major concerns with this paper in its current form.

Reviewer #2: In the presented manuscript, Anand et al. authors performed a phylogenetic analysis of SASR-COV-2 variants to determine the most conserved region of the virus and performed a virtual screening against determined proteins FDA-approved drugs. Further, they performed molecular docking and determined the drugs Darifenacin, Nebivolol, Bictegravir, Alvimopan, and Irbesartan as the most promising candidates.

In the first part of the paper, the phylogenetic analyses are unnecessary and misleading. There is a number of high-quality publications (e.g. doi.org/10.1101/2020.04.17.046086 ., DOI: 10.1002/jmv.25902, DOI: 10.1002/jmv.25699) and publicly available resource for phylogenetic analysis of SARS-COV-2 variants https://nextstrain.org/. Therefore, to avoid controversial and unsupported conclusions of coronavirus's potential origin and transmission roads, I recommend focusing on virtual screening and docking of the FDA-approved drugs, presented in the second part of the manuscript.

With the modifications, the paper could be suitable for publication

Response:

We understand the reviewer’s concern. We have removed our phylogenetic analysis from the first part of the manuscript. We have paid more attention towards the virtual screening, docking and MD simulations of the repurposed FDA approved drugs in the manuscript. Instead of 25 ns simulations that were initially done, the MD simulations are now extended to an optimal time of 100 ns. The time of 100 ns was chosen since the RMSD of the protein-drug complexes stabilized in the early phases of the simulation. Respective changes have been made under the “Molecular Dynamic simulations” in the results and discussion section and in the “Materials and Methods” section. The figures for 100 ns MD simulations have been added in the “Molecular Dynamic simulations” section. The various parameters that were considered in the MD analysis for each protein have been listed in a separate table (Table 4).

Reviewer #3: This paper presents a very standard approach to both the phylogenetic analysis of SARS-CoV2 and the homology modeling building and virtual ligand screening. There are a number of issues that preclude publication in the following form:

1. The paper is disjoint. What insights does the phylogenetic analysis provide with respect to the drug repurposing aspect of the algorithm?

Response 1:

We have removed our phylogenetic analysis from the first part of the manuscript and have edited the manuscript in order to bring coherence to the story. We have assigned more weight to the conserved gene approach and drug targeting in the current manuscript.

2. There are numerous technical and conceptual problems with the ligand screening/homology modeling. 25 ns or so simulations are too short to see significant structural changes. How were the the particular drugs chosen? What is the point of doing a Lipinski analysis on FDA approved drugs?

Response 2: 

We have increased the simulation timings to 100 ns in order to observe structural details at a finer level. The time of 100 ns was chosen since the RMSD of the protein-drug complexes stabilized in the early phases of the simulation. During the screening of ligands for each protein, the drugs from the FDA approved database were docked to the protein and the PLANTS software provided a docking score for each of the drugs. For every docked protein, the top two drug candidates with higher docking score were chosen and were moved for the further MD analysis. We have removed the Lipinski analysis part on the repurposed drugs.

3. The paper targets nsp10. But nsp10 forms a complex with nsp16 which is likely to be a biologically relevant conformation. How does the dimer affect the putative ligand binding with nsp10? Why not simulate the dimer since there is a crystal structure of the nsp10/nsp16 complex?

Response 3:

We agree with the comments raised by the reviewer. We have reanalysed the screening, docking and the MD simulation with NSP10-NSP16 complex by using the crystal structure of the complex available in the PDB database with RCSB PDB ID 7C2I. Indeed there was a change in the ligand-binding pocket when the NSP10-NSP16 complex was considered. While the binding of Alvimopan with the complex was stable, the NSP-Irbesartan complex was not stable as evident from the MD analysis. Since both NSP10 and NSP10-NSP16 complex have different preferred ligand-binding sites, we have also decided to retain the simulations that were originally performed with NSP10. We have modified the figures and provided the results of the new analysis.

4. What is the false positive rate of the docking approach that is used? How confident are the binding predictions?

Response 4: 

We have used PLANTS approach to perform our docking analysis. The tool provides scores which proxies the success rate for each protein-ligand docking. The confidence of these binding predictions for the dockings done in the manuscript has been added in the “Homology modelling of stable targets and virtual screening of small molecules” in the results and discussion section. The rationale behind using the PLANTS approach is explained in the materials and methods section.

5. The authors need to perform experimental validation that their predictions are correct.

Response 5:

In order to validate the results from the haplotype analysis, homology modelling, virtual screening and molecular docking, we have performed ab-initio molecular dynamics simulations. The time duration of production MD was chosen to make sure that all the docked complexes were indeed stable. The RMSD of the docked complexes stabilized well before 40ns. However, we continued to simulate the systems for another 60ns to verify the stability of the docked systems.

In addition to computational experiments, several existing published articles indicate that the selected repurposed drugs are effective in combating the viral infection. Some existing literature evidence supporting our study are listed below:

1. Zhou Y, Hou Y, Shen J, Huang Y, Martin W, Cheng F. Network-based drug repurposing for novel coronavirus 2019-nCoV/SARS-CoV-2. Cell Discovery. 2020;6(1).

2. Kim J, Zhang J, Cha Y, Kolitz S, Funt J, Escalante Chong R et al. Advanced bioinformatics rapidly identifies existing therapeutics for patients with coronavirus disease-2019 (COVID-19). Journal of Translational Medicine. 2020;18(1).

3. Guan W, Lan W, Zhang J, Zhao S, Ou J, Wu X et al. COVID-19: Antiviral Agents, Antibody Development and Traditional Chinese Medicine. Virologica Sinica. 2020;35(6):685-698.

4. Akhtar S, Benter I, Danjuma M, Doi S, Hasan S, Habib A. Pharmacotherapy in COVID-19 patients: a review of ACE2-raising drugs and their clinical safety. Journal of Drug Targeting. 2020;28(7-8):683-699.

5. Chatterjee B, Thakur S. ACE2 as a potential therapeutic target for pandemic COVID-19. RSC Advances. 2020;10(65):39808-39813. 

6. Enmozhi S, Raja K, Sebastine I, Joseph J. Andrographolide as a potential inhibitor of SARS-CoV-2 main protease: an in silico approach. Journal of Biomolecular Structure and Dynamics. 2020;:1-7.

7. Bocci G, Bradfute S, Ye C, Garcia M, Parvathareddy J, Reichard W et al. Virtual and In Vitro Antiviral Screening Revive Therapeutic Drugs for COVID-19. ACS Pharmacology & Translational Science. 2020;3(6):1278-1292.

Overall, this paper represents an incremental contribution to a very important and difficult problem.

---

## [Decision Letter · Decision Letter 1]

23 Feb 2021

PONE-D-20-34651R1

A comprehensive SARS-CoV-2 genomic analysis identifies potential targets for drug repurposing

PLOS ONE

Dear Dr. Pradhan,

Thank you for submitting your manuscript to PLOS ONE. After careful consideration, we feel that it has merit but does not fully meet PLOS ONE’s publication criteria as it currently stands. Therefore, we invite you to submit a revised version of the manuscript that addresses the points raised during the review process.

I went over your responses to the queries to the reviewers and agree that they have been addressed appropriately. The manuscript is basically acceptable.

However, I have the following additional suggestions to improve the palatability of the manuscript to broader reader base.  There is no  major change to the methodology, or interpretation of the results, or conclusions. 

@Abstract:

Plpro and 3CLpro : please use their common names NSP## in addition, and mark the second names in figure 1.

Please remove "a novel approach for drug targeting". This is the standard approach to design/develop new drugs.

Please remove " This approach of targeting the stable genes for drug discovery would provide a better therapeutic approach and confidence in the successive clinical trials. We also identified the

mutation frequencies across the viral genome". This is only a projection, and the mutation frequencies are used from other more comprehensive mutation tracking databases.

@heading :Viral clusters identified via haplotype network

Please add a statement at the end of this paragraph that, this haplotype network may be incomplete because of the origin of the sequences from specific regions. This limitation is intended to reflect that there might be more haplotypes in other parts of the world which are under-reported due to limited access to sequencing. Can you also add a rough date here representing the time of this analysis? Also in materials and methods. This will help the manuscript serve as a guide when the mutation landscape changes in future and someone tries to repeat the steps.

@"Nucleotide positions 240, 3036, 8781, 11082, 14407, 23402, 28143, with reference to NC_045512"

Please add the version number as NC_045512.2. And ensure that these numbers match to this specific version. The previous version had ~30 bases less, so the positions would be incorrect if those were the positions at the time of study. No correction needed if these numbers already come from the version 2.

@results section. I notice that the findings are arranged by the methods followed by the genes and drugs.

Root Mean Square Deviation (RMSD)

The radius of Gyration (Rg)

Intermolecular Hydrogen Bonding

Root Mean Square Fluctuations (RMSF)

Solvent Accessible Surface Area analysis (SASA)

While this approach is fine for those who have a deeper understanding over these methodologies, it is hard for broader reader-base to collate the information. I suggest you to re-arrange them by the gene/protein; then in each, one drug at a time, then describe all the finer aspects above (RMSD, Rg, IHB, RMSF, and SASA). There is no need to change the figures or numbering. That way, the first interaction would be more comprehensive with some brief information about the methods, then the remaining would be easier for the reader because of the same pattern of description. It also helps if someone wants to focus a specific gene or drug.

I believe this is increase the palatability of the manuscript..

We look forward to receiving your revised manuscript.

Kind regards,

Malaya Kumar Sahoo, Ph.D

Academic Editor

PLOS ONE

Journal Requirements:

Reviewers' comments:

Reviewer's Responses to Questions

**Comments to the Author**

1. If the authors have adequately addressed your comments raised in a previous round of review and you feel that this manuscript is now acceptable for publication, you may indicate that here to bypass the “Comments to the Author” section, enter your conflict of interest statement in the “Confidential to Editor” section, and submit your "Accept" recommendation.

Reviewer #1: All comments have been addressed

2. Is the manuscript technically sound, and do the data support the conclusions?

Reviewer #1: Yes

3. Has the statistical analysis been performed appropriately and rigorously? 

Reviewer #1: I Don't Know

4. Have the authors made all data underlying the findings in their manuscript fully available?

Reviewer #1: Yes

5. Is the manuscript presented in an intelligible fashion and written in standard English?

Reviewer #1: Yes

6. Review Comments to the Author

Reviewer #1: I was satisfied with the corrections they have made. I have no major concerns with this paper in its current form.

7. PLOS authors have the option to publish the peer review history of their article (what does this mean?). If published, this will include your full peer review and any attached files.

Reviewer #1: **Yes: **raja sab kalluru

---

## [Author Response · Author response to Decision Letter 1]

27 Feb 2021

Dear Dr. Malaya Kumar Sahoo,

Academic Editor

PLOS ONE

We thank you for the precise comments on our manuscript [PONE-D-20-34651] - [EMID:e297d3cb56d8414a] entitled “A Comprehensive SARS-CoV-2 Genomic Analysis Identifies Potential Targets for Drug Repurposing”. We have gone through your comments carefully, and have addressed them in this revised version of the manuscript. We have modified the paper in response to the remarks, and hope that this revised version of the manuscript is suitable for publication in the journal.

We have appended a detailed response in the following pages.

Thank you.

Best wishes,

Arpit Kumar Pradhan

LMU Munich

Ashwin K. Jainarayanan

University of Oxford

Response to Editor’s comments:

@Abstract:

Plpro and 3CLpro : please use their common names NSP## in addition, and mark the second names in figure 1.

Please remove "a novel approach for drug targeting". This is the standard approach to design/develop new drugs.

Please remove " This approach of targeting the stable genes for drug discovery would provide a better therapeutic approach and confidence in the successive clinical trials. We also identified the

mutation frequencies across the viral genome". This is only a projection, and the mutation frequencies are used from other more comprehensive mutation tracking databases.

Response: The common names for the respective proteases as well as their NSP names have been added. Further, we have made the required changes as were suggested by the editor. 

@heading :Viral clusters identified via haplotype network

Please add a statement at the end of this paragraph that, this haplotype network may be incomplete because of the origin of the sequences from specific regions. This limitation is intended to reflect that there might be more haplotypes in other parts of the world which are under-reported due to limited access to sequencing. Can you also add a rough date here representing the time of this analysis? Also in materials and methods. This will help the manuscript serve as a guide when the mutation landscape changes in future and someone tries to repeat the steps.

Response: We have made the required changes as suggested by the editor.

@"Nucleotide positions 240, 3036, 8781, 11082, 14407, 23402, 28143, with reference to NC_045512"

Please add the version number as NC_045512.2. And ensure that these numbers match to this specific version. The previous version had ~30 bases less, so the positions would be incorrect if those were the positions at the time of study. No correction needed if these numbers already come from the version 2.

Response: We went through both the reference versions (NC_045512 and NC_045512.2) and the nucleotide positions are in reference to version number NC_045512.2. The corresponding changes, have been made in the manuscript. 

@results section. I notice that the findings are arranged by the methods followed by the genes and drugs.

Root Mean Square Deviation (RMSD)

The radius of Gyration (Rg)

Intermolecular Hydrogen Bonding

Root Mean Square Fluctuations (RMSF)

Solvent Accessible Surface Area analysis (SASA)

While this approach is fine for those who have a deeper understanding over these methodologies, it is hard for broader reader-base to collate the information. I suggest you to re-arrange them by the gene/protein; then in each, one drug at a time, then describe all the finer aspects above (RMSD, Rg, IHB, RMSF, and SASA). There is no need to change the figures or numbering. That way, the first interaction would be more comprehensive with some brief information about the methods, then the remaining would be easier for the reader because of the same pattern of description. It also helps if someone wants to focus a specific gene or drug.

I believe this is increase the palatability of the manuscript.

Response: The MD section in the results has been rearranged with the parameters being described under individual proteins.

---

## [Editor Report · Decision Letter 2]

2 Mar 2021

A comprehensive SARS-CoV-2 genomic analysis identifies potential targets for drug repurposing

PONE-D-20-34651R2

Dear Dr. Pradhan,

We’re pleased to inform you that your manuscript has been judged scientifically suitable for publication and will be formally accepted for publication once it meets all outstanding technical requirements.

Kind regards,

Malaya Kumar Sahoo, Ph.D

Academic Editor

PLOS ONE
---

## [Editor Report · Acceptance letter]

8 Mar 2021

PONE-D-20-34651R2 

A comprehensive SARS-CoV-2 genomic analysis identifies potential targets for drug repurposing 

Dear Dr. Pradhan:

I'm pleased to inform you that your manuscript has been deemed suitable for publication in PLOS ONE. Congratulations! Your manuscript is now with our production department. 

Kind regards, 

on behalf of

Dr. Malaya Kumar Sahoo 

Academic Editor

PLOS ONE